# Naloxegol for the Treatment of Opioid-Induced Constipation in Patients with Cancer Pain: A Pooled Analysis of Real-World Data

**DOI:** 10.3390/cancers17050865

**Published:** 2025-03-03

**Authors:** Jean-Marc Sabaté, Carmen Beato-Zambrano, Manuel Cobo, Antoine Lemaire, Vincenzo Montesarchio, Judith Serna-Montros, Rafik Namane, Santiago Martín Baccarelli, Fernando Rico-Villademoros

**Affiliations:** 1Gastroenterology and Gastrointestinal Oncology, Hôpital Avicenne, AP-HP, Sorbonne University, 93000 Bobigny, France; 2INSERM U987, Pathophysiology and Clinical Pharmacology of Pain, 92012 Boulogne Billancourt, France; 3Medical Oncology, Hospital Universitario Virgen Macarena, 41009 Sevilla, Spain; cbeatoz@hotmail.com; 4Medical Oncology, Instituto de Investigación Biomédica de Málaga-Plataforma BIONAND (IBIMA-BIONAND), Hospital Regional Universitario Malaga, 29010 Malaga, Spain; manuelcobodols@yahoo.es; 5Oncology & Medical Specialties Department, Valenciennes General Hospital, 59300 Valenciennes, France; lemaire-a@ch-valenciennes.fr; 6Pneumology and Oncology, A.O.R.N. dei Colli-Monaldi Hospital, 80131 Napoli, Italy; vincenzo.montesarchio@ospedalideicolli.it; 7Palliative Care/Medical Oncology Vhio, Hospital Campus Vall d’Hebron, 08035 Barcelona, Spain; judith.serna@vallhebron.cat; 8Laboratoires Grünenthal, 92800 Puteaux, France; rafik.namane@grunenthal.com; 9Data Management & Biostatistics Department, APICES, Pinto, 28320 Madrid, Spain; santiago.martin@apices.es; 10Medical Department, APICES, Pinto, 28320 Madrid, Spain; fernando.rico-villademoros@apices.es

**Keywords:** naloxegol, PAMORA, opioid-induced constipation, cancer pain, real world, pooled analysis

## Abstract

Opioid medications relieve cancer pain, but they also cause side effects, such as constipation. Results from clinical trials conducted to grant marketing authorization may differ from those observed in clinical practice. Therefore, data from those trials should be complemented with information from studies conducted in the real-world practice settings. In this study, we combined the results of three studies conducted with naloxegol under clinical practice conditions. We found that naloxegol improved opioid-induced constipation and quality of life in patients with cancer pain. These results, together with those of clinical trials in patients with non-cancer pain, support the use of naloxegol for the management of opioid-induced constipation in patients with cancer pain.

## 1. Introduction

Pain is a common symptom of patients with cancer that frequently requires treatment with opioids [1]. Constipation is the most frequent and bothersome side effect in patients receiving opioid treatment [2], affecting over 50% of patients with cancer pain [3]. Opioid-induced constipation (OIC) has a similar symptomatology as functional constipation, and it is associated with considerable psychosocial and economic burden [4]. Importantly, the occurrence of OIC impacts treatment with opioids in terms of reducing the dose of opioids or even discontinuing treatment, which in turn frequently results in inadequate pain control [2,5].

OIC is frequently underdiagnosed and especially undertreated [6,7]. In an analysis of 216 patient-provider discussions in patients receiving opioids for noncancer pain, 139 patients reported constipation. Of these patients, in 105 (75%), physicians considered that could be OIC, and in 47 (34%), no specific action was recommended [8]. This situation could reflect a discordance in the perception of the importance and severity of OIC between patients and healthcare providers [9]. In patients with cancer pain, there are additional barriers to the management of OIC, such as the prioritization of other symptoms (e.g., pain, diarrhea, anxiety) or treatment toxicities over constipation [10].

In patients with OIC, clinical practice guidelines recommend traditional laxatives (i.e., osmotic, stimulant, or detergent/surfactant stool softeners) as the first-line treatment [11]. However, laxatives frequently fail to relieve constipation in OIC patients and are also associated with side effects that can interfere with patients’ work and social activities [2,12,13,14]. In patients with laxative-refractory OIC, the use of peripherally acting µ-opioid receptor antagonists (PAMORAs) is recommended [11]. PAMORAs are a class of drugs indicated for OIC that specifically bind and block the µ-opioid receptors of the gastrointestinal tract without crossing the blood–brain barrier; thus, they may alleviate OIC without interfering with the analgesic effect of opioids [15,16].

Naloxegol is a pegylated derivative of naloxone with increased oral bioavailability and enhanced peripheral selectivity and was, after methylnaltrexone, the second PAMORA approved by the Food and Drug Administration (FDA) for the management of OIC in adult patients with chronic noncancer pain [11]. The efficacy and safety of naloxegol in OIC in patients with noncancer pain were demonstrated in two 12-week, placebo-controlled, randomized clinical trials [17]. An additional 52-week, open-label, randomized trial comparing naloxegol 25 mg with usual care in patients with noncancer pain and OIC revealed that naloxegol was generally safe and well tolerated [18].

Although randomized controlled trials are the gold standard for evaluating the effect and value of an intervention due to their internal validity, they have limitations in establishing the benefits of treatment in clinical practice because the characteristics of patients and the provision of treatment usually differ from those of randomized controlled trials [19]. Moreover, recruiting patients with advanced cancer for a randomized clinical trial can be challenging not only because of disease severity but also because of the reluctance of family members who perceive the patient as too ill to participate in a clinical trial [20]. Therefore, data from randomized controlled trials should be complemented with information from real-world practice settings [21]. The aim of the NALOPOOL project was to assess the efficacy and safety of naloxegol in patients with cancer pain who exhibited OIC and were treated under real-world practice conditions. To this end, we pooled data from three observational studies with naloxegol conducted in that population [22,23,24].

## 2. Materials and Methods

This was a pooled analysis of three observational prospective studies [22,23,24] comprising 427 patients with cancer pain and OIC treated with naloxegol. All studies complied with ethical and local regulations on observational studies, including the provision of informed consent to every patient. The main characteristics of the studies are shown in Appendix A and summarized below.

### 2.1. Study Design and Patients

All three studies were prospective and observational studies conducted in medical oncology, palliative care, and pain departments/units. While the Kyonal and Move studies were multicenter studies conducted in a single country (Spain and France, respectively), the Nacasy study was a multicenter study conducted in several European countries. The study duration was 4 weeks in the Move and Nacasy studies [23,24] and 12 months in the Kyonal study [22]. Regardless of the study duration, only 4 weeks of data were pooled in this analysis because it was the study duration for the Move and Nacasy studies.

The studies included adult patients with cancer pain requiring treatment with opioids and who experienced OIC. With slight differences across studies, the OIC definitions were compatible with the Rome IV criteria [25]. Although the Kyonal and Move studies explicitly included patients who did not respond to laxatives, this inclusion criterion does not appear among the inclusion criteria of the Nacasy study. Nonstringent exclusion criteria were applied, with slight differences among the studies. The Kyonal study explicitly excluded patients with cognitive impairment [22], the Move study excluded those with gastrointestinal obstruction [23], and the Nacasy study excluded those diagnosed with colorectal cancer [24]; however, gastrointestinal obstruction is a contraindication included in the naloxegol’s summary of product characteristics and, therefore, should be considered an exclusion criterion for all three studies.

### 2.2. Study Assessments

Regarding the efficacy measures, only bowel movements recorded in the patients’ diaries and the Patient Assessment of Constipation Quality-of-Life Questionnaire (PAC-QOL) were recorded or administered in the three studies and were considered the key efficacy measures in this pooled analysis.

The PAC-QOL is a 28-item self-reported measurement of the burden of constipation on patients’ everyday functioning and well-being in the 2 weeks (14 days) prior to assessment that was developed in parallel with the Patient Assessment of Constipation Symptoms (PAC-SYM) [26]. It includes four subscales (worries and concerns, physical discomfort, psychosocial discomfort, and satisfaction) and an overall scale. Lower scores indicate better quality of life.

Other efficacy measures that were administered in only two out of the three studies included the PAC-SYM questionnaire (administered in the Kyonal and Move studies), the Bowel Function Index (BFI) (administered in the Move and Nacasy studies), and the Bristol Stool Scale (BSS) (administered in the Kyonal and Nacasy studies).

The PAC-SYM questionnaire is a 12-item self-administered questionnaire that evaluates the severity of symptoms of constipation in three domains (stool, rectal, and abdominal symptoms), with lower scores indicating a lower symptom burden [27,28]. The BFI is a physician-administered three-item questionnaire (ease of defecation, feeling of incomplete bowel evacuation, and personal judgment of constipation) specifically developed and validated to assess the severity of constipation-associated symptoms in individuals with OIC, with lower scores indicating lower symptom severity [29]. The BSS evaluates stool consistency at the time of every bowel movement, categorizing stools into seven types ranging from type 1 (hard lumps) to type 7 (watery diarrhea) [30].

With respect to tolerability/safety measures, adverse events were recorded in the Move and Nacasy studies, whereas adverse reactions were recorded in the Kyonal study. Therefore, in this pooled analysis, we considered only adverse reactions (i.e., adverse events assessed as at least possibly related to the study drug). For this analysis, adverse reactions were coded with the Medical Dictionary for Regulatory Activities (MedDRA).

### 2.3. Statistical Analysis

Key efficacy outcomes included the response at week 4, defined as three or more bowel movements at week 4 with an increase of one or more bowel movements over baseline (in the Nacasy study, only the former part of the definition had to be met), a mean change from baseline in the total and subscale scores of the PAC-QOL at week 4, and the proportion of responders in the quality-of-life questionnaire at week 4, defined as those with a mean change in the PAC-QOL score at week 4 of 0.5 or greater.

Other efficacy measures included the impact of treatment on constipation symptoms as evaluated with the PAC-SYM and the BFI using the following outcomes:Mean change from baseline in the PAC-SYM total score and subscores at week 4.% of responders, according to the PAC-SYM, are defined as those with a mean change in the total score of 0.5 or greater at week 4.Mean change from baseline in the BFI total score and subscores at week 4.% of responders in the BFI, defined as those with a reduction of ≥12 points in the BFI total score at week 4.% of patients with a BFI score of <30 at week 4, which is considered well-controlled OIC.

Finally, the impact on stool consistency was evaluated with the mean change from baseline in the BSS score at week 4.

The efficacy population consisted of patients who met selection criteria, had received at least one dose of naloxegol, and had undergone the post-baseline efficacy assessment. Safety analyses comprised all patients who met all selection criteria and had received at least one dose of the study drug.

Categorical variables are described using absolute and relative frequencies. Continuous variables are summarized as the means and standard deviations. To compare scores between visits, we used the Student’s *t*-test for paired samples or ANOVA, as applicable. In addition, for the mean changes from baseline in the scale scores, Cohen’s d was calculated and interpreted as small, moderate, and large changes for effect sizes of 0.2–0.49, 0.50–0.79, and >0.80, respectively. Binary outcomes are presented as percentages and corresponding 95% confidence intervals. No imputation of values for missing data was performed, and we used a visit-wise approach. Heterogeneity was explored using Cochran’s Q-test or Levene’s test. An exploratory logistic regression analysis for evaluating the factors associated with treatment response was performed via stepwise forward automatic Wald entry and included study as a fixed factor and the following covariates: age, sex, body mass index, current chemotherapy, presence of metastases, opioid treatment duration (categorized as a binary outcome using the median [<8.9 months vs. ≥8.9 months]), laxative use and naloxegol starting dose (12.5 mg vs. 25 mg).

All analyses were performed with SPSS version 26 (IBM Corp., Armonk, NY, USA) statistical software.

## 3. Results

### 3.1. Patient Disposition and Characteristics

Overall, 442 patients were included in the three studies, and 427 were included in this pooled analysis; 393 were analyzed for baseline characteristics and included in the efficacy analyses, and 427 were included in the tolerability and safety analyses (Figure 1).

The baseline characteristics differed across the studies (Table 1). Compared with those in the other two studies, patients in the Nacasy study were older, had a slight predominance of females, were less likely to have received chemotherapy at the time of study entry, had a shorter opioid treatment duration, and were less likely to have received laxatives at the time of study entry (i.e., patients had received previous laxative treatment in 68% of the cases in the Nacasy study compared to 94% and 99% of the patients in the Kyonal and Move studies, respectively). The type of opioid treatment also differed across the studies, with fentanyl (73%) being the most common opioid in the Kyonal study, oxycodone (82%) in the Move study, and fentanyl, oxycodone, and morphine (received by 29%, 27% and 14% of the patients, respectively) in the Nacasy study.

The initial dose of naloxegol was 25 mg in most patients (n = 325, 82.7%). Changes in opioid dose were recorded only in the Move and Nacasy studies, with 43 (16.1%) of the pooled 267 patients showing a dose increase, 17 (6.4%) showing a dose reduction, and 2 (0.7%) discontinuing opioid treatment.

### 3.2. Key Efficacy Outcomes

The pooled proportion of responders was 71.0% (95% CI, 65.9–75.9), with homogeneous results (Figure 2). In the multivariate analysis, the single factor associated with the occurrence of treatment response was body mass index as a continuous variable (odds ratio 1.03, 95% CI 1.01–1.05, *p* < 0.001).

The mean changes from baseline in the PAC-QOL subscores and overall scores at week 4 were statistically significant and clinically relevant, with moderate effect sizes for the ‘psychosocial discomfort’ and ‘worries and concerns’ dimensions and large effect sizes for the ‘physical discomfort’ and ‘satisfaction’ dimensions and the overall score (Table 2; Appendix A). The pooled proportion of patients with clinically relevant improvement in the PAC-QOL was 59.9% (95% 54.3% to 65.5%), with homogenous results across the studies (Figure 3).

### 3.3. Impact on Constipation Symptoms and Stool Consistency

The mean changes from baseline in terms of constipation symptoms to week 4, as evaluated with the PAC-SYM and BFI scores, are presented in Table 3 (see also Appendix A). Changes in the total scores and subscores were statistically significant, with large effect sizes in the PAC-SYM and BFI total scores and some dimensions of the PAC-QOL (‘stool symptoms’) and the BFI (‘ease of defecation’ and ‘self-judgment of constipation’). Significant heterogeneity was found in the pooled analyses of the PAC-SYM dimensions but not in the analysis of the total score.

The pooled proportion of patients with a clinically relevant change in constipation symptoms was 68.9% (95% CI 62.3% to 73.5%; homogeneity Cochran’s Q = 0.01, *p* = 0.930), as evaluated with the PAC-SYM, and 68.2% (95% CI 61.7% to 74.7%; homogeneity Cochran’s Q = 1.70, *p* = 0.190), as evaluated with the BFI. The pooled proportion of patients with a BFI score <30 at week 4 was 35.5% (95% CI 28.9% to 42.1%; homogeneity Cochran’s Q = 0.20, *p* = 0.660).

Stool consistency, as evaluated with the BSS, significantly improved by week 4, with a moderate effect size (mean change from baseline 1.0 [95% CI 0.78 to 1.22], Cohen’s d = 0.56), but the results were heterogeneous (Levene’s homogeneity test *p* < 0.001).

### 3.4. Safety and Tolerability

The pooled proportion of patients who discontinued the drug due to adverse reactions was 6.1% (95% CI 3.8% to 8.4%; homogeneity Cochran’s Q = 0.72, *p* = 0.700). Three patients experienced serious adverse reactions: two cases of diarrhea and one case of intestinal perforation. The case of intestinal perforation occurred in a 68 years old male who was diagnosed with advanced pancreatic cancer and had a medical history of gastric bypass surgery; this patient is described in greater detail in the original report of the Nacasy study [24]. One patient reported a withdrawal syndrome of mild severity.

Adverse reactions by severity could only be pooled for the Move and Nacasy studies and are presented in Table 4. The most frequent adverse reactions were abdominal pain (n = 15, 5.0%), diarrhea (n = 10, 3.3%), and nausea (n = 3, 1.0%); most adverse reactions were Grade 1 or 2. In the Kyonal study, the adverse reactions were abdominal pain (n = 9, 7.1%), abdominal distension (n = 5, 4.0%), diarrhea (n = 4, 3.2%), nausea (n = 3, 2.4%) and dysesthesia (n = 1, 0.8%); most adverse reactions were categorized as mild (n = 17) or moderate (n = 3) in severity.

## 4. Discussion

The results of this pooled analysis of three observational studies evaluating the use of naloxegol in cancer pain patients with OIC revealed that 4 weeks of treatment with naloxegol was associated with a response in terms of bowel movements in more than two-thirds of the patients. Other efficacy measures for evaluating constipation symptoms have shown consistent results, with relevant improvements also observed in more than two-thirds of the patients and a relevant improvement in the quality of life in 60% of the patients. Naloxegol was generally well tolerated, with a tolerability profile consistent with that reported in pivotal studies.

The response rate in this pooled analysis was 71%, which was higher than that reported in two pivotal randomized studies in noncancer patients with OIC: 44.4% in the KODIAC-04 study and 39.7% in the KODIAC-05 study [17]. Several factors could explain these differences. First, patients in the pivotal studies are likely to be more chronic, with a mean duration of opioid use of 41–49 months, whereas the mean duration of opioid treatment in this analysis was approximately 6 months. Second, the use of concomitant laxatives and other treatments differed between the KODIAC studies and the Nalopool study. In the KODIAC studies, 66% to 72% of the patients had used laxatives within the previous two weeks, and in this pooled analysis, 86% of the patients were receiving laxatives at baseline. Notably, in the KODIAC study, most prior laxatives were stimulants, whereas the use of stimulant laxatives was low in the studies included in our pooled analysis. However, it is important to note that this pattern of use was homogenous across the three studies; therefore, it is likely to reflect a real-world pattern of use in our setting. In addition, in this pooled analysis, 53% of the patients received chemotherapy, which, as a secondary side effect, could impact this outcome. Third, the more conservative intent-to-treat analysis of the pivotal studies and the longer duration of treatment (12 weeks vs. 4 weeks) could be associated with a reduced treatment effect compared with the visit-wise approach performed in our analysis. Finally, the definition of response in the KODIAC studies was different than in these observational studies; thus, the primary endpoint was the 12-week response rate (more than or equal to three spontaneous bowel movements per week and an increase from baseline of more than or equal to one spontaneous bowel movements for ≥9 of 12 weeks and for ≥3 of the final 4 weeks). The results with other PAMORAs in this population are scarce, but our results are in line with those of two real-world studies and one randomized clinical trial with naldemedine. Hiruta et al. [31], in a retrospective study conducted in Japan, reported that 203 of 255 (79.6%) evaluable patients with cancer pain and OIC experienced at least three bowel movements per week at the study endpoint after treatment with naldemedine. Using a similar definition as in our analysis, Nishiba et al. [32] reported, in another retrospective Japanese study, a response rate of 65.7% after 7 days of treatment with naldemedine. Finally, in a phase III, 2-week, randomized, double-blind, placebo-controlled trial, Katakami et al. [33] reported an SBM response rate of 71.1%.

In our exploratory multivariate analysis, the single factor associated with treatment response was BMI; for each point increase in BMI, the likelihood of response increased by 3%. In the Move study, the factors associated with response were the presence of bone metastases (OR 3.62) and a shorter duration of opioid treatment (OR 5.12); it is unclear whether BMI was included in the analyses of the Move study [23]. In a retrospective study with naldemedine in this population, the single factor associated with response was a dose of opioids less than 30 mg of morphine equivalent; although the effect of BMI on response showed the same direction as in our study, it was not statistically significant (a BMI < 22 compared with a BMI ≥ 22 had an OR of 0.77, *p* = 0.51). Previous studies in other settings evaluating the association between BMI and constipation have shown mixed results [34,35]. In our view, BMI may be a proxy for other factors that are associated with treatment response, such as better health status; however, a higher BMI is associated with a sedentary lifestyle, a factor associated with constipation. Overall, we do not have a robust explanation for this association.

The results of the secondary efficacy measures were consistent with those of the response rate, with statistically significant and clinically relevant improvements in all dimensions of constipation symptoms and in stool consistency. A clinically relevant improvement in constipation symptoms was observed in ≈68% of the patients, as evaluated with the PAC-SYM and BFI. Furthermore, more than one-third of the patients had a BFI score <30 at week 4, which is below the threshold selected by experts to define the need for prescription medication in patients with OIC [36].

OIC is associated with low self-esteem and feelings of embarrassment and significantly interferes with daily life activities and productivity [4]. This impact on quality of life is important, such that for many patients with OIC, it is challenging to balance the need for adequate pain relief and the occurrence of constipation [37]. In patients with advanced cancer, this issue is even more important since the goal of palliative care is to improve the quality of life of patients [38]. Our results show that treatment with naloxegol improved quality of life to a clinically relevant extent. Importantly, all dimensions of quality of life showed statistically significant and clinically relevant improvements. Despite the heterogeneity of the populations of patients with OIC included in the three studies of this pooled analysis, the results of the improvement in quality of life overlap, with a proportion of patients showing a clinically relevant improvement in quality of life that ranged from 58.7% in the Move study to 61.1% in the Kyonal study.

There were no new tolerability or safety issues arising from these studies. The adverse reactions were consistent with the tolerability profile of naloxegol and other PAMORAs, with abdominal pain and diarrhea as the most frequent adverse reactions. Most of the adverse reactions were mild to moderate (e.g., 11 of the 15 cases of abdominal pain and 9 of the 10 cases of diarrhea reported in the Move and Nacasy studies). There was a single case of drug withdrawal syndrome that was categorized as mild. There were three serious adverse reactions, including a case of intestinal perforation, which eventually resulted in death.

Our study has several limitations. As is frequently the case in pooled analyses, we included only observational studies sponsored by the manufacturer of naloxegol. However, we are aware of only one other observational study of naloxegol in patients with cancer [39]; this study does not provide information on response rates, but the PAC-QOL and BFI results are consistent with our results. Although most of our pooled analyses did not show heterogeneity, this finding should be interpreted with caution because of the small number of studies included in our analyses. The closer monitoring of patients in a randomized controlled than in an observational study trial could explain the low frequency of adverse reactions in our pooled analyses. In contrast, the major strength of these observational studies and pooled analyses is that we provide information on the efficacy and tolerability of naloxegol in a population that is otherwise difficult to include and study in randomized clinical trials [20]. Thus, in a multicenter, randomized, placebo-controlled study designed to assess the feasibility of a definitive trial of naloxegol for OIC in patients with cancer, of 590 patients who were pre-screened, only 4 patients were randomized, and the study was closed after 24 months because of poor recruitment [20]. Finally, it would have been interesting to perform a pooled analysis of all observational studies with PAMORAs, but that implies having individual patient data for all studies, and usually, that is not possible.

## 5. Conclusions

Despite the above limitations, our results suggest that in the clinical practice setting, treatment of OIC in patients with cancer pain with naloxegol is consistently effective in more than two-thirds of patients, is associated with relevant improvement in quality of life, and has a tolerability and safety profile consistent with that reported in noncancer patients. These results, together with those of randomized clinical trials in patients with noncancer pain, support the use of naloxegol for the management of OIC in patients with cancer pain who do not respond to treatment with laxatives.

## Figures and Tables

**Figure 1 cancers-17-00865-f001:**
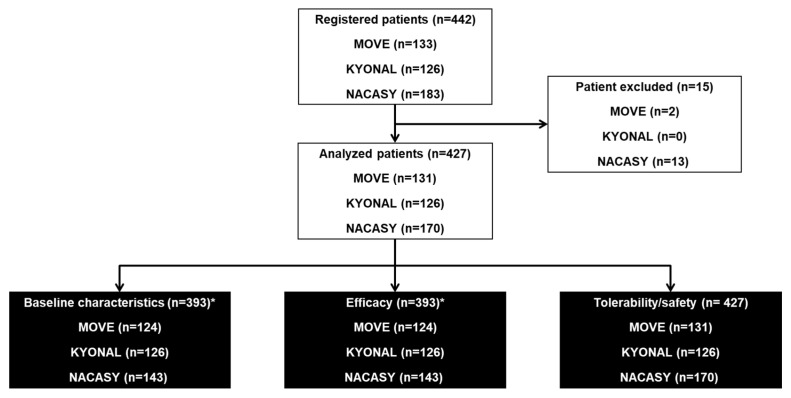
Patient disposition. Patients were excluded because they did not meet the inclusion criteria (n = 13), and treatment with naloxegol was not initiated. * 34 patients were excluded from the efficacy analysis: 7 patients from the Move study (2 because of absence of cancer and 5 because they had not received laxatives for a minimum of four days; 27 patients from the Nacasy study because of lack of a post-baseline visit.

**Figure 2 cancers-17-00865-f002:**
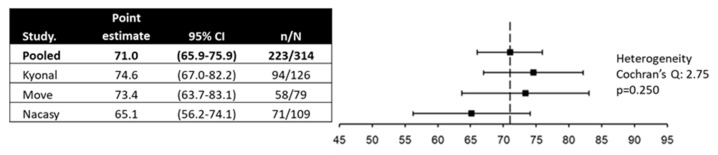
Response rate over the 4-week treatment period.

**Figure 3 cancers-17-00865-f003:**
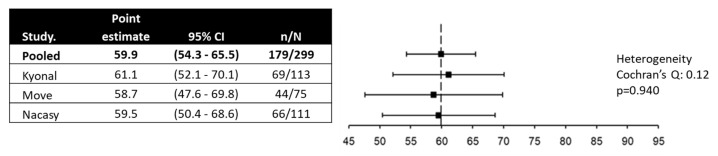
The proportion of patients with clinically relevant improvement in quality of life.

**Table 1 cancers-17-00865-t001:** Baseline characteristics.

Baseline Characteristics	Kyonal N = 126	Move N = 124	Nacasy N = 143	Total N = 393
Age, years, mean (SD)	61.5 (12.2)	62.1 (12.1)	64.1 (12.4)	62.7 (12.2)
Body mass index, mean (SD)	25.0 (4.6)	23.8 (4.7)	25.0 (5.3)	24.6 (4.9)
Sex, female, N (%)	52 (41.3)	55 (44.4)	78 (54.5)	185 (47.7)
Primary tumor location, N (%)				
Lung	45 (35.7)	17 (13.7)	35 (24.5)	97 (24.7)
Breast	21 (16.7)	19 (15.3)	32 (22.4)	72. (18.3)
Prostate	11 (8.7)	13 (10.5)	12 (8.4)	36 (9.2)
Head and neck	11 (8.7)	6 (4.8)	8 (5.6)	25 (6.4)
Bladder	4 (3.2)	8 (4.8)	6 (4.2)	16 (4.1)
Pancreas	0 (0.0)	3 (2.4)	10 (7.0)	13 (3.3)
Digestive	13 (10.3)	0 (0.0)	0 (0.0)	13 (10.3)
Kidney	0 (0.0)	8 (6.5)	3 (2.1)	11 (2.8)
Presence of metastasis, yes, N (%)	85 (67.5)	75 (60.5)	99 (69.2)	259 (65.9)
Current chemotherapy treatment, yes, N (%)	77 (61.1)	67 (54.0)	66 (46.2)	210 (53.4)
Opioid treatment at baseline, N (%) *				
Fentanyl	92 (73.0)	36 (29.0)	41 (28.7)	169 (43.0)
Oxycodone	15 (11.9)	101 (81.5)	38 (26.6)	154 (39.2)
Morphine	33 (26.2)	55 (44.4)	20 (14.0)	108 (27.5)
Tramadol	2 (1.6)	9 (7.3)	8 (5.6)	19 (4.8)
Methadone	4 (3.2)	1 (0.8)	10 (7.0)	15 (3.8)
Hydromorphone	0 (0.0)	3 (2.4)	10 (7.0)	13 (3.3)
Codeine	0 (0.0)	2 (1.6)	11 (7.7)	13 (3.3)
Oxycodone/Naloxone	0 (0.0)	0 (0.0)	10 (7.0)	10 (2.5)
Opioid treatment duration (weeks), mean (SD)	23.6 (33.9)	34.7 (59.7)	16.8 (33.7)	24.9 (44.5)
Previous laxative treatment, yes, N (%)	118 (93.7)	123 (99.2)	97 (67.8)	338 (86.0)
Previous laxative treatments, N (%) **				
Osmotic	106 (89.8)	116 (94.3)	85 (87.6)	307 (90.8)
Enema	6 (5.1)	19 (15.4)	11 (11.3)	36 (10.7)
Stimulant	14 (11.9)	3 (2.4)	50 (14.8)	33 (9.8)
Stool softeners	9 (7.6)	2 (1.6)	12 (12.4)	23 (6.8)

* Patients could receive more than one treatment. ** Figures are calculated over those receiving laxatives. SD, standard deviation.

**Table 2 cancers-17-00865-t002:** Mean changes from baseline in the Patient Assessment of Constipation Quality-of-Life Questionnaire scores.

	Kyonal	Move	Nacasy	Pooled	Cohen’s d	Heterogeneity
PAC-QoL—Physical Discomfort	0.9 (0.9)	0.8 (0.9)	0.9 (1.0)	0.9 (1.0)	0.91	0.379
PAC-QoL—Psychosocial Discomfort	0.6 (0.9)	0.5 (0.9)	0.8 (0.9)	0.6 (0.9)	0.69	0.391
PAC-QoL—Worries and Concerns	0.8 (0.9)	0.6 (0.9)	0.8 (1.0)	0.7 (0.9)	0.78	0.478
PAC-QoL—Satisfaction	0.8 (0.7)	0.7 (0.7)	0.6 (0.7)	0.7 (0.7)	0.91	0.193
PAC-QoL—Global	0.7 (0.8)	0.6 (0.7)	0.8 (0.8)	0.7 (0.8)	0.93	0.063

Unless otherwise indicated, the figures represent the mean change (standard deviation). All changes were statistically significant at *p* < 0.0001. PAC-QOL, Patient Assessment of Constipation Quality-of-Life Questionnaire.

**Table 3 cancers-17-00865-t003:** Mean changes (SD) from baseline in constipation symptoms and stool consistency.

	**Kyonal**	**Move**	**Nacasy**	**Pooled**	**Cohen’s d**	**Heterogeneity**
**Constipation symptoms**						
PAC-SYM—Abdominal	0.6 (0.9)	0.8 (1.0)	NR	0.7 (0.9)	0.70	0.007
PAC-SYM—Rectal	0.7 (0.9)	0.6 (1.0)	NR	0.7 (0.9)	0.74	0.015
PAC-SYM—Stool	1.2 (1.2)	1.1 (1.0)	NR	1.1 (1.1)	1.04	0.015
PAC-SYM—Total	0.8 (0.8)	0.8 (0.7)	NR	0.8 (0.8)	1.03	0.622
BFI—Ease of defecation	NR	27.4 (33.8)	28.4 (32.3)	28.0 (32.8)	0.85	0.801
BFI—Feeling incomplete evacuation	NR	25.8 (33.6)	24.3 (37.0)	24.9 (35.6)	0.70	0.418
BFI—Self-judgment of constipation	NR	39.8 (33.7)	32.2 (34.8)	35.3 (34.5)	1.02	0.322
BFI—Total score	NR	30.9 (29.9)	28.1 (31.5)	29.2 (30.8)	0.95	0.677
**Stool consistency**						
BSS score	1.4 (1.3)	NR	0.5 (2.1)	1.0 (1.7)	0.57	<0.001

Unless otherwise indicated, the figures represent the mean change (standard deviation). All changes were statistically significant at *p* < 0.0001, except for those of the ’abdominal symptoms’ and ’rectal symptoms’ dimensions of the PAC-SYM, which were statistically significant at *p* = 0.001. BFI, Bowel Function Index; BSS, Bristol Stool Scale; NR, not reported; PAC-SYM, Patient Assessment of Constipation Symptoms.

**Table 4 cancers-17-00865-t004:** Adverse reactions pooled for the Move and Nacasy studies.

	Grade											
	1		2		3		5		NA		Total	
Preferred Term	N	%	N	%	N	%	N	%	N	%	N	%
Abdominal pain	4	1.3	7	2.3	1	0.3	0	0.0	3	1.0	15	5.0
Diarrhea	4	1.3	5	1.7	0	0.0	0	0.0	1	0.3	10	3.3
Nausea	1	0.3	2	0.7	0	0.0	0	0.0	0	0.0	3	1.0
Flatulence	0	0.0	1	0.3	1	0.3	0	0.0	0	0.0	2	0.7
Constipation	0	0.0	0	0.0	0	0.0	0	0.0	1	0.3	1	0.3
Eructation	1	0.3	0	0.0	0	0.0	0	0.0	0	0.0	1	0.3
Gastrointestinal pain	1	0.3	0	0.0	0	0.0	0	0.0	0	0.0	1	0.3
Intestinal perforation	0	0.0	0	0.0	0	0.0	1	0.3	0	0.0	1	0.3
Vomiting	0	0.0	1	0.3	0	0.0	0	0.0	0	0.0	1	0.3
Drug withdrawal syndrome	1	0.3	0	0.0	0	0.0	0	0.0	0	0.0	1	0.3
Fatigue	0	0.0	1	0.3	0	0.0	0	0.0	0	0.0	1	0.3
Pain	0	0.0	1	0.3	0	0.0	0	0.0	0	0.0	1	0.3
Withdrawal syndrome	1	0.3	0	0.0	0	0.0	0	0.0	0	0.0	1	0.3
Decreased appetite	0	0.0	1	0.3	0	0.0	0	0.0	0	0.0	1	0.3
Pollakiuria	1	0.3	0	0.0	0	0.0	0	0.0	0	0.0	1	0.3

The intensity of the adverse reactions was evaluated with the National Cancer Institute Common Terminology Criteria with the following grades: Grade 1 = mild; Grade 2 = moderate; Grade 3 = severe or medically significant but not immediately life-threatening; Grade 4 = life-threatening consequences; and Grade 5 = death related to the AE.

## Data Availability

Data will be available on request due to restrictions (privacy or ethics).

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
