# Peer review of "Naloxegol for the Treatment of Opioid-Induced Constipation in Patients with Cancer Pain: A Pooled Analysis of Real-World Data"

_cancers, 2025, doi:10.3390/cancers17050865_

Round 1
Reviewer 1 Report
Comments and Suggestions for Authors
Personally, I took on a difficult task in undertaking this review, as I am not a practicing clinician researcher, but a preclinical opioid researcher. For this reason, I have no detailed knowledge of the scales used in the article or the statistical methods used for these scales. Since it would greatly aid understanding, I suggest that the various questionnaires indicated in the study be included as supplementary material to the article. It would be important to show in the tables presenting the results not only the changes in scores but also the absolute scores at the beginning and end of the period, so that the extent of the change would be more apparent, especially to a reader who is not familiar with the subject. Nevertheless, the information presented in the manuscript is understandable and important for practical purposes.
I would suggest a few minor corrections:
1. line 75 states that PAMORs inhibit opioid receptors in the GI tract, but this is not entirely true. These compounds inhibit the entire peripheral mu opioid pool. However, this is of little practical relevance until a peripherally acting opioid analgesic is available.
2. Line 79 states that the first PAMORA approved for the treatment of non-cancer patients was nalexegol. This is probably true, but in general methyl-naltrexone was the first such drug. This should be noted for completeness, especially as the present study involved cancer patients.
3. in Table 1, the total number of participants should be included in the first row, although this is included later, but should be in-documented here. Also in this table, it is observed that probably due to combination treatment or drug switching, but the percentage of different opioids in total exceeds 100%. This should be briefly described. Also, it is strange to me that in the Nacasy study 10 patients received an oxycodone/naloxone combination, as in this formulation naloxone acts as a PAMORA, blocking GI tract receptors, so I think this should have been an exclusion criterion from the outset. It is not clear exactly why stimulants and bisacodyl are listed separately in the laxatives, as bisacodyl is probably the most commonly used stimulant.
4. For Figure 1, I would like to ask why the number of elements is different for the efficacy analysis and the tolerability analysis. Is it due to treatment discontinuation? Probably a typo: in the caption of the figure for exclusion it says n=13 but it is n=15.
5. In the case of Figures 2 and 3, it is not clear to me why the table header shows point estimate instead of number of respondents?
6. As I wrote in my introduction, it would help understanding in Tables 2 and 3 if we could see not only the changes but also the absolute numbers of the initial and final values. If not here, at least as supplementary material.
7. It is reported that BMI was the only factor that showed a correlation with efficiency. What other factors were examined?
8. I do not understand the statement in lines 335-336. Does this mean that the treatment is still not effective enough? Or, how important is this despite the treatment available?
Author Response
We thank the reviewer for taking the time to review and comment on our manuscript. We found the advice constructive and have incorporated most suggestions in our review.
Reviewer 1
Personally, I took on a difficult task in undertaking this review, as I am not a practicing clinician researcher, but a preclinical opioid researcher. For this reason, I have no detailed knowledge of the scales used in the article or the statistical methods used for these scales. Since it would greatly aid understanding, I suggest that the various questionnaires indicated in the study be included as supplementary material to the article.
We agree that this would be a good addition to the manuscript. However, owing to copyright issues, we could not include the questionnaire exhibits.
It would be important to show in the tables presenting the results not only the changes in scores but also the absolute scores at the beginning and end of the period, so that the extent of the change would be more apparent, especially to a reader who is not familiar with the subject. Nevertheless, the information presented in the manuscript is understandable and important for practical purposes.
Including this information in the tables in the manuscript would complicate this information. However, as suggested, we have included information on the absolute scores in the Supplementary Information (supplementary tables 2-14). The extent of the change could also be evaluated using the effect size that appears in the tables.
I would suggest a few minor corrections:
- line 75 states that PAMORs inhibit opioid receptors in the GI tract, but this is not entirely true. These compounds inhibit the entire peripheral mu opioid pool. However, this is of little practical relevance until a peripherally acting opioid analgesic is available.
We understand from your comment that no modifications are required.
- Line 79 states that the first PAMORA approved for the treatment of non-cancer patients was nalexegol. This is probably true, but in general methyl-naltrexone was the first such drug. This should be noted for completeness, especially as the present study involved cancer patients.
We have changed the sentence accordingly and currently reads (Lines 79-80):
“Naloxegol is a pegylated derivative of naloxone with increased oral bioavailability and enhanced peripheral selectivity and was, after methylnaltrexone, the second PAMORA approved by the Food and Drug Administration (FDA) for the management of OIC in adult patients with chronic noncancer pain”
- in Table 1, the total number of participants should be included in the first row, although this is included later, but should be in-documented here. Also in this table, it is observed that probably due to combination treatment or drug switching, but the percentage of different opioids in total exceeds 100%. This should be briefly described
Based on your suggestion, we have included the number of participants in Table 1. Regarding the number of opioids, we have included a footnote clarifying that patients could receive more than one treatment.
Also, it is strange to me that in the Nacasy study 10 patients received an oxycodone/naloxone combination, as in this formulation naloxone acts as a PAMORA, blocking GI tract receptors, so I think this should have been an exclusion criterion from the outset.
We acknowledge the reviewer’s concern regarding the inclusion of 10 patients who received oxycodone/naloxone in the Nacasy study. As noted, naloxone in this formulation acts as a peripherally acting μ-opioid receptor antagonist (PAMORA) (even if, strictly speaking, it does not belong to this class due to the possibility of central effects in case of high doses or liver failure).
However, oxycodone/naloxone is a medicine primarily indicated for the management of pain. Thus, for instance, in Australia the indication reads: “The management of moderate to severe chronic pain unresponsive to non-narcotic analgesia. The naloxone component in a fixed combination with oxycodone is indicated for the therapy and/or prophylaxis of opioid-induced
For real-world studies, especially observational studies, we believe that their inclusion is adequate.
It is not clear exactly why stimulants and bisacodyl are listed separately in the laxatives, as bisacodyl is probably the most commonly used stimulant.
The reviewer correctly points out that bisacodyl is a stimulant laxative, and its separate listing may create redundancy. The original categorization followed the structure reported in the individual studies included in our pooled analysis.
However, we agree that merging bisacodyl under the "stimulant laxatives" category would enhance clarity and reduce potential confusion. Therefore, we have changed this information in the table collapsing the two categories
- For Figure 1, I would like to ask why the number of elements is different for the efficacy analysis and the tolerability analysis. Is it due to treatment discontinuation? Probably a typo: in the caption of the figure for exclusion it says n=13 but it is n=15.
This difference is due to the different definitions of the efficacy and safety populations for the analyses. However, these definitions were not included in the manuscript. We have added the following information in the Methods section (Lines 176-179):
“Efficacy population consisted of patients who met selection criteria, had received at least one dose of naloxegol and had the post-baseline efficacy assessment. Safety analyses comprised all patients who met all selection criteria and had received at least 1 dose of the study drug”.
- In the case of Figures 2 and 3, it is not clear to me why the table header shows point estimate instead of number of respondents?
What we are presenting in the figure is the frequency (%). The numbers of respondents (numerator and denominator) also appear in the table.
- As I wrote in my introduction, it would help understanding in Tables 2 and 3 if we could see not only the changes but also the absolute numbers of the initial and final values. If not here, at least as supplementary material.
Including this information in the tables in the manuscript would complicate this information. However, as suggested, we have included information on the absolute scores in the Supplementary Information. The extent of the change could also be evaluated using the effect size that appears in the tables.
- It is reported that BMI was the only factor that showed a correlation with efficiency. What other factors were examined?
The factors included in the model are already described in Methods. Thus, it currently reads (Lines 188-193):
“An exploratory logistic regression analysis for evaluating the factors associated with treatment response was performed via stepwise forward automatic Wald entry and included study as a fixed factor and the following covariates: age, sex, body mass index, current chemotherapy, presence of metastases, opioid treatment duration (categorized as a binary outcome using the median [<8.9 months vs. ≥8.9 months]), laxative use and naloxegol starting dose (12.5 mg vs. 25 mg)”.
- I do not understand the statement in lines 335-336. Does this mean that the treatment is still not effective enough? Or, how important is this despite the treatment available?
We mean that a substantial proportion of patients met a very stringent efficacy criterion. To clarify this issue, we have reworded the sentence. Now it reads (Lines 349-351):
“Furthermore, more than one-third of the patients had a BFI score <30 at week 4, which is below the threshold selected by experts to define the need for prescription medication in patients with OIC”
Reviewer 2 Report
Comments and Suggestions for Authors
The abstract needs to include some major missing points, especially that the patients had cancer pain and constipation (mostly) not responding to laxatives.
The second paragraph describes bowel care standards that many centres might consider poor, and readers have to go to the references to see where and in what context these observations were made. It is very difficult to know what the pre-naolxegol laxative therapy was. It would be helpful to have the referenced studies’ basic information provided here. Who was managing the bowel care? Oncologists? General Practitioners? Nurses etc? What was meant by an inadequate response to laxatives? It wasn’t even an inclusion criterion for one of the three contributing trials. In another it had only to be for 4 days, which is not sufficient to declare constipation unresponsive. If not titrated to effect in a structured protocol and given time, then OIC cannot be considered actually refractory to standard care.
Prior laxative use is somewhat unclear in the text and Table 1. Previous laxative therapy is reported in 67.8% in one study, 89.6 and 94.3 in the two other studies, 86% overall, which leaves 36.2- 5.7% of the subjects NOT on a prior laxative, whereas the conclusion states that naloxegol is beneficial when there has been be inadequate response to laxative therapy. That may well be true, but this study compilation cannot clearly state that without this information. The use of stimulant laxatives also seems substantially less than would be the norm in most places, and may explain the presumed ineffectiveness of laxative therapy. Docusate is also still on this list of prior laxatives, whereas there is good evidence that it is ineffective and has been dropped from most bowel protocols for some years now, so should not be included as a laxative.
The care setting of the patients included is not described: were these ambulatory patients or hospitalized?
Comments on the Quality of English Language
There are some awkward sentences, most likely due to imperfect translation, and the paper would benefit from review for readability.
Author Response
We thank the reviewer for taking the time to review and comment on our manuscript. We found the advice constructive and have incorporated most suggestions in our review.
Reviewer 2
The abstract needs to include some major missing points, especially that the patients had cancer pain and constipation (mostly) not responding to laxatives.
We have modified the abstract accordingly. Thus, the first sentence of the methods section in the abstract reads (Lines 34-35):
“We pooled individual patient data from three multicenter observational studies conducted with naloxegol in patients with cancer who exhibited OIC and were prescribed naloxegol under real-world conditions”.
The second paragraph describes bowel care standards that many centres might consider poor, and readers have to go to the references to see where and in what context these observations were made. It is very difficult to know what the pre-naolxegol laxative therapy was. It would be helpful to have the referenced studies’ basic information provided here. Who was managing the bowel care? Oncologists? General Practitioners? Nurses etc? What was meant by an inadequate response to laxatives? It wasn’t even an inclusion criterion for one of the three contributing trials. In another it had only to be for 4 days, which is not sufficient to declare constipation unresponsive. If not titrated to effect in a structured protocol and given time, then OIC cannot be considered actually refractory to standard care.
In addition to Supplementary Table 1, to clarify the issues posed by the reviewer, we have included the following information in the Methods section.
Lines 110-111 - “All three studies were prospective and observational studies conducted in medical oncology, palliative care, and pain departments/units”.
Lines 119-121 - “Although the Kyonal and Move studies explicitly included patients who did not respond to laxatives, this inclusion criterion does not appear among the inclusion criteria of the Nacasy study”.
Please note that we do not state that nalexegol is efficacious in patients with OIC refractory to laxatives. We only took into account the indication nalexegol in Europe: “Moventig is indicated for the treatment of opioid-induced constipation (OIC) in adult patients who have had an inadequate response to laxative
Prior laxative use is somewhat unclear in the text and Table 1. Previous laxative therapy is reported in 67.8% in one study, 89.6 and 94.3 in the two other studies, 86% overall, which leaves 36.2- 5.7% of the subjects NOT on a prior laxative, whereas the conclusion states that naloxegol is beneficial when there has been be inadequate response to laxative therapy. That may well be true, but this study compilation cannot clearly state that without this information.
This fact is already recognized in the Results section when describing the differences across the study populations. Thus, we state the following:
“The baseline characteristics differed across the studies (Table 1). Compared with those in the other two studies, patients in the Nacasy study were older, had a slight predominance of females, were less likely to have received chemotherapy at the time of study entry, had a shorter opioid treatment duration, and were less likely to have received laxatives at the time of study entry”.
To further clarify that issue, we have modified second sentence as follows (lines 204-210):
“The baseline characteristics differed across the studies (Table 1). Compared with those in the other two studies, patients in the Nacasy study were older, had a slight predominance of females, were less likely to have received chemotherapy at the time of study entry, had a shorter opioid treatment duration, and were less likely to have received laxatives at the time of study entry (i.e. Patients had received previous laxative treatment in 68% of the cases in the Nacasy study compared to 94% and 99% of the patients in the Kyonal and Move studies, respectively”)”
Currently our conclusions are read (verbatim):
“Despite the above limitations, our results suggest that in the clinical practice setting, treatment of OIC in patients with cancer pain with naloxegol is consistently effective in more than two-thirds of patients, is associated with relevant improvement in quality of life, and has a tolerability and safety profile consistent with that reported in noncancer patients. These results, together with those of randomized clinical trials in patients with noncancer pain, support the use of naloxegol for the management of OIC in patients with cancer pain who do not respond to treatment with laxatives”.
Therefore, we believe that this conclusion is consistent with the design and results of this study. The last sentence of the conclusion (“who do not respond to treatment with laxatives”) was written in that way to be consistent with summary of product of characteristics (see above).
The use of stimulant laxatives also seems substantially less than would be the norm in most places, and may explain the presumed ineffectiveness of laxative therapy. Docusate is also still on this list of prior laxatives, whereas there is good evidence that it is ineffective and has been dropped from most bowel protocols for some years now, so should not be included as a laxative.
We appreciate the reviewer’s insight regarding the underuse of stimulant laxatives in our dataset. The relatively lower use of stimulant laxatives in our study population compared to clinical norms may indeed contribute to the perception of laxative inefficacy, as stimulant laxatives are generally considered more effective for OIC than osmotic laxatives alone.
This discrepancy likely reflects regional differences in prescribing habits and the real-world nature of our pooled data, which includes patients from various healthcare settings. We have clarified this point in the discussion to acknowledge that laxative use patterns may influence treatment response rates. Thus, when discussing the response rates and differences with the KODIAC studies, we have added the following sentences (Lines 310-313):
“Notably, in the KODIAC study, most prior laxatives were stimulants, whereas the use of stimulant laxatives was low in the studies included in our pooled analysis. However, it is important to note that this pattern of use was homogenous across the three studies; therefore, it is likely to reflect a real-world pattern of use in our setting”.
The care setting of the patients included is not described: were these ambulatory patients or hospitalized?
We have included the information provided by the articles in the Methods section, which is limited to the specialties involved in the management of patients included in the studies (see a previous comment). Unfortunately, the publications do not provide information on whether the patients were ambulatory or hospitalized.
Comments on the Quality of English Language
There are some awkward sentences, most likely due to imperfect translation, and the paper would benefit from review for readability.
The manuscript has been edited by a professional editing company. We have attached the certificate.

Round 2
Reviewer 2 Report
Comments and Suggestions for Authors
I am happy with the changes and author responses.